# Electron-Beam Welding of Titanium and Ti6Al4V Alloy

Georgi Kotlarski [1], Darina Kaisheva [1,2,*], Maria Ormanova [1], Borislav Stoyanov [3], Vladimir Dunchev [4], Angel Anchev [4] and Stefan Valkov [1,5]

1   Institute of Electronics, Bulgarian Academy of Sciences, 72 Tzarigradsko Chausse Blvd, 1784 Sofia, Bulgaria; gvkotlaraki@ie.bas.bg (G.K.); m.ormanova@ie.bas.bg (M.O.); stvalkov@ie.bas.bg (S.V.)
2   Department of Physics, South-West University "Neofit Rilski", 66 Ivan Michailov Str., 2700 Blagoevgrad, Bulgaria
3   Department of Industrial Design and Textile Engineering, Technical University of Gabrovo, 4 H. Dimitar Str., 5300 Gabrovo, Bulgaria; b.stoyanov@tugab.bg
4   Department of Material Science and Mechanics of Materials, Technical University of Gabrovo, 4 H. Dimitar Str., 5300 Gabrovo, Bulgaria; v.dunchev@tugab.bg (V.D.); anchev@tugab.bg (A.A.)
5   Department of Mathematics, Informatics, and Natural Sciences, Technical University of Garbovo, 4 H. Dimitar Str., 5300 Gabrovo, Bulgaria
*   Correspondence: darinakaisheva@ie.bas.bg

**Abstract:** This work presents the results of the electron-beam welding of commercially pure α-Ti (CP-Ti) and Ti6Al4V (Ti64) alloys. The structure and mechanical properties of the formed welded joints were examined as a function of the power of the electron beam. The beam power was set to $P_1 = 2100$ W, $P_2 = 1500$ W, and $P_3 = 900$ W, respectively. X-ray diffraction (XRD) experiments were performed in order to investigate the phase composition of the fabricated welded joints. The microstructure was examined by both optical microscopy, scanning electron microscopy (SEM), and energy dispersive X-ray spectroscopy (EDX). The mechanical properties of the formed joints were studied using tensile test experiments and microhardness experiments. The results of the experiments were discussed concerning the influence of the beam power on the microstructure and the mechanical properties of the weld joints. Furthermore, the practical applicability of the present method for the welding of α-Ti and Ti64 was also discussed.

**Keywords:** electron beam welding; titanium; Ti6Al4V alloy; structure; mechanical properties

## 1. Introduction

The intensive study of titanium and its alloys for a variety of applications in many industries and areas of human life began in the middle of the 20th century. Titanium has a low density of about 4.50 g/cm³, which is much lower than that of steel, at 8.03 g/cm³. At the same time, its mechanical characteristics, especially the high temperature wear resistance, are comparable to those of martensitic steels. Titanium has exceptional corrosion resistance, even higher than that of stainless steel in a variety of aggressive high corrosion environments, and exhibits excellent corrosion resistance and bio-compatibility in the human body. Titanium can be cast, forged, and easily processed by various technologies [1]. In addition, the price of titanium is lower than that of high-grade steel based alloys such as Kovar and Inconel. All of these properties determine the widespread use of titanium and its alloys in the chemical [2], automotive [3,4], shipbuilding [5], aerospace [6–8], and sports equipment [9] industries, as well as in the medical field for implementation in orthopedic implants [10–12] or in orthodontic implants [13–15].

The CP-Ti (commercially pure titanium) and the Ti64 (Ti6Al4V) alloy are materials with slightly different thermo-physical properties due to the presence of alloying elements in the structure of Ti64. Despite having similar thermal and electrical properties, they have vastly different mechanical properties. Ti6Al4V has significantly higher yield strength, ultimate tensile strength, shear strength, and higher hardness. However, it characterizes as

brittle with about 10–15% of elongation. Titanium, in comparison, has poorer mechanical properties, although it possesses higher plasticity and slightly better thermal conductivity. The welding of titanium and titanium alloy can be realized via gas tungsten arc welding (GTAW), gas metal arc welding (GMAW), plasma arc welding (PAW), laser beam welding (LBW), and electron beam welding (EBW) [16]. GTAW, GMAW, and PAW have the advantage of a large weld seam, which is advantages in the case of the welding of thick work pieces. However, during the welding process, while using these methods a large heat affected zone is observed, which can lead to a change in the structure and mechanical properties of the welded materials. The advantage of LBW and EBW in that regard is the narrow size of the heat sources, which means that a higher precision welding is possible. Furthermore, the welded materials are less thermally affected during the welding process due to the smaller imprint of the laser/electron beams. These methods are thus able to be used for the welding of thin specimens with complex geometries.

In industrial fields, electron beam welding is somewhat problematic due to the larger size of electron beam units compared to laser welding equipment. In addition, the size of the work pieces is also limited to the volume of the vacuum chamber. However, the utilization of vacuum systems during the process of welding is exactly what makes electron beam welding attractive. The high vacuum environment guarantees the high quality of welding, with no adsorption of gases in the weld seam. Furthermore, the vacuum environment slows the process of cooling, thus reducing the formed internal stresses between the weld pool and the substrates. This means that in using this technology it is possible to weld dissimilar metals such as aluminum and copper [17], copper and stainless steel [18], etc.

Among the published works are those that address issues regarding the welding of only Ti [19] or only Ti64 [20–23], as well as the dissimilar welding of different titanium alloys such as VT9 [24,25], Ti5Al2.5Sn [26], and TC4 [27,28]. There is almost no research devoted to the EBW of Ti and Ti64 in the literature. As mentioned, these materials have different mechanical properties which in all cases can be problematic and lead to the formation of defects in the structure of the weld seam and the reduction of its mechanical properties. Therefore, the current work focuses on welding between Ti and Ti64.

In this study, the possibility of welding CP-Ti and Ti6Al4V using an electron beam as a heat source was presented. The influence of the beam power on the structure and mechanical characteristics of the welded joints was studied. The results present insight on the further development of the method for its further optimization.

## 2. Materials and Methods

As mentioned previously, the studied materials are commercially pure titanium (CP-Ti) and Ti-6Al-4V (Ti64). The chemical composition in wt.% of the untreated Ti64 substrates is as follows: 5.8% Al, 4.67% V, 0.16% Fe, 0.09% Co, 0.07% Mo, 0.02% Pd, 0.14% Hf, bal. Ti. The substrates were in the form of welding plates with a size of 100 mm × 50 mm × 8 mm. EBW was carried out on the Evobeam Cube 400 welding unit manufactured by Evobeam using the following technological conditions: welding speed v = 10 mm/s; accelerating voltage U = 60 kV; beam current $I_1 = 35$ mA, $I_2 = 25$ mA, and $I_3 = 15$ mA. The corresponding beam power was $P_1 = 2100$ W, $P_2 = 1500$ W, and $P_3 = 900$ W, respectively. The scheme of the EBW of CP-Ti and Ti64 plates is presented in Figure 1.

Scanning electron microscopy (SEM) was performed on a "LYRA I XMU" scanning electron microscope for the investigation of the microstructure of the welded specimens. This analysis was performed in order to gain knowledge on the resultant microstructure as a function of the used beam power. All images included in this paper were taken using the back-scattered electrons mode. The accelerating voltage during the experiments was 20 kV. In addition, energy dispersive X-ray spectroscopy was performed in order to obtain information on the chemical composition of the studied samples.

A Bruker D8 Advance X-ray diffractometer was used to analyze the investigated samples' phase. The method used was Coupled Two Theta. The radiation of the X-ray tube was CoKα with a wavelength 1.78897 Å. The range of the research was from 35° to 115°,

the X-ray generator current was 40 mA, and the used voltage was 35 kV. The tests were performed with a 0.05° step with a 0.25 s time for each step.

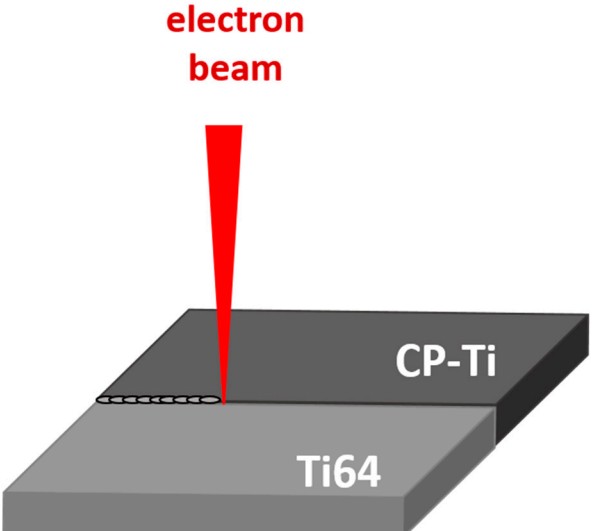

**Figure 1.** Schematic of the process of EBW of CP-Ti and Ti64.

The tensile tests were performed on a machine for static and dynamic tests (a ZWICK Vibrophore 100) in accordance with the requirements of the ISO 6892-1 Method B, with a constant rate of stresses in the elastic region of 30 MPa/s. The microhardness measurements were performed in a semi-automatic mode. A visual representation of the method is shown in Figure 2. During the experiments, 40 measurement of the Vickers hardness were performed. For each measurement a force of 0.5 N was employed along with a test time of 10 s.

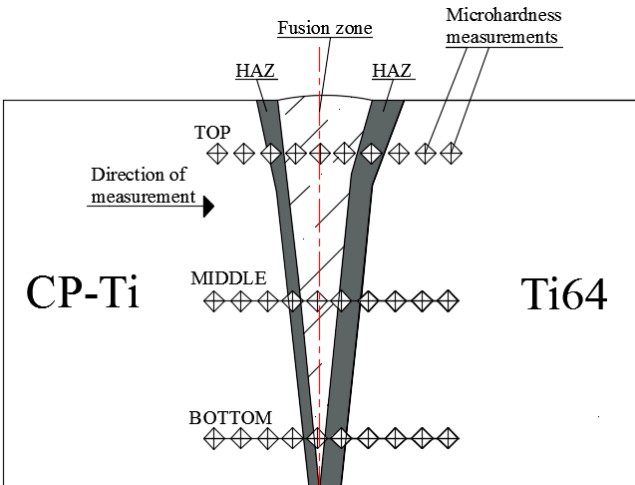

**Figure 2.** Scheme of the Vickers hardness experiments.

### 3. Results

In order to study the basic morphology of the welded specimens, metallographic samples of the cross-section of each specimen were prepared. Figure 3 shows the resultant cross-sections of the weld seams as well as the fusion zones (FZ) and the heat affected zones (HAZ). The sample shown in Figure 3a corresponds to the cross-section of the specimens welded using the highest used power of the heat source: 2100 W, the sample shown in Figure 3b to the one cross-section of the specimens welded using the lower beam power of 1500 W, and the third one (Figure 3c) shows the cross-section of the specimens welded

using a power of 900 W. Evidently, the full penetration of the electron beam through the height of the specimens was achieved in the first two cases, and in the case of the lowest beam power only half penetration was observed. This indicates that the beam power of 900 W is insufficient to reach the full penetration of the 8 mm thick specimens. Due to the shallow penetration achieved in that case, any further study of the structure and mechanical properties of that sample was discontinued. The much lower cross-section of the weld which formed at a beam power of 900 W compared to the ones formed at a beam power of 1500 W and 2100 W indicates that during tensile testing no comparable data between the samples could be obtained. This means that no clear relationship between the microstructure and the mechanical properties of that sample can be established. In the case of the specimens welded using a beam power of 2100 W and 1500 W, a noticeable difference in the geometry of the fusion zone and the heat affected zones was observed. The specimen shown in Figure 3b has the traditional welding keyhole shape [29] of the formed weld associated with electron beams. The heat-affected zones follow the geometry of the keyhole weld almost ideally. In Figure 3a, both the fusion zone and the heat-affected zones have a rectangular shape, indicating that the power of the electron beam in this case was sufficiently high to easily penetrate the substrates without causing the formation of a particular shape of the weld pool. The formation of such a narrow weld seam leads to a subsequent reduction in the size of the heat affected zones, and thus the lower spread of the thermal energy along the length of the specimens.

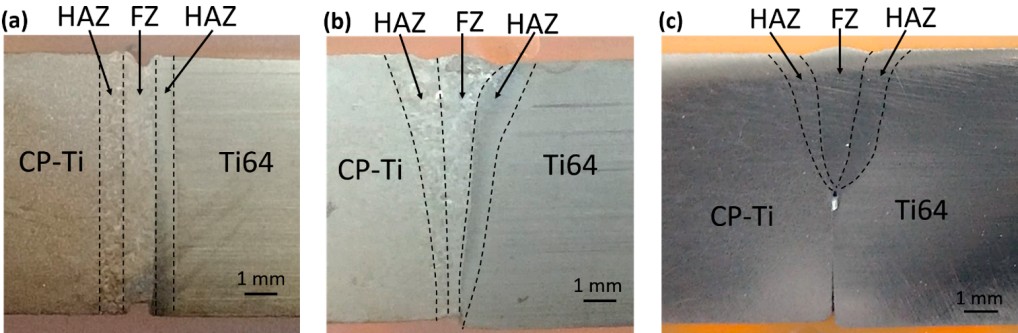

**Figure 3.** Optical microscope images of the weld seam formed using a beam power of: (**a**) 2100 W, (**b**) 1500 W, and (**c**) 900 W.

Figure 4 shows scanning electron microscopy images of the untreated substrates. Figure 4a shows the structure of the CP-Ti substrate, where only $\alpha$-Ti grains were observed. Figure 4b shows the structure of the Ti64 welding plate, which indeed consists of both a mixed $\alpha$-Ti and $\beta$-Ti grains.

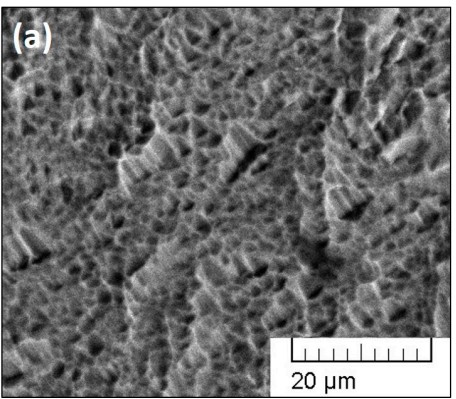
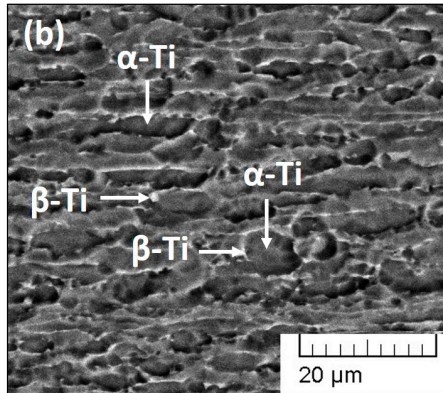

**Figure 4.** SEM images of the untreated materials: (**a**) CP-Ti, and (**b**) Ti64.

Figure 5 presents SEM images of the fusion zone of the two studied specimens—sample 1, welded with a beam power of 2100 W (Figure 5a), and sample 2, welded with a power

of 1500 W (Figure 5b). The obtained images indicate that the first fusion zone formed during the welding of the specimens at a beam power of 2100 W consists primarily of pure α-Ti particles with some inclusions of the α'-Ti martensitic phase. A transformation of the α-Ti phase into the α'-Ti martensitic phase was observed, which led to a refinement of the structure of the weld seam. The results compliment the results obtained during the X-ray diffraction analysis.

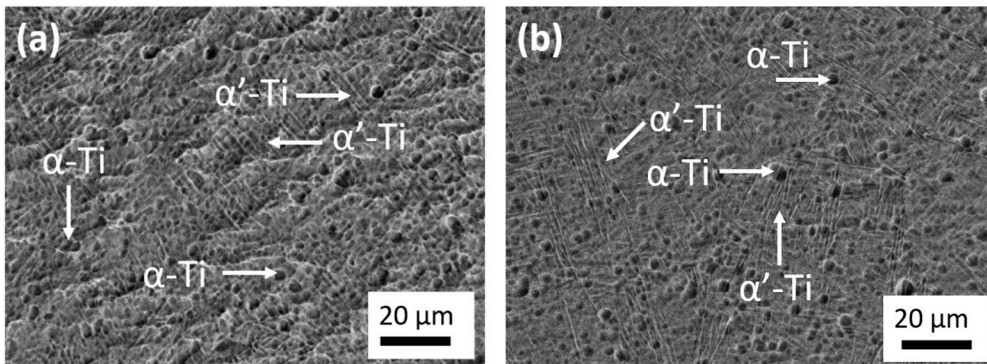

**Figure 5.** SEM images of the fusion zone of the specimens welded with a power of the electron beam of: (**a**) 2100 W, (**b**) 1500 W.

The phase composition of all weld seams is presented in Figure 6, along with the original phase composition of the substrate materials used as controls. In the case of the pure titanium substrate peaks, a clear α-Ti phase with a hexagonal closed-packed structure was observed. Such a diffractogram is standard for pure grade titanium substrates [30]. In the case of Ti64, a similar structure was observed, with the exception of the additional β-Ti phase which demonstrates a body-centered cubic structure. Vanadium in the structure of the Ti6Al4V alloy is used as a beta-stabilizing element, which forms the excellent balance of that alloy between the standard α-Ti phase and the secondary β-Ti phase, and which results in the improved properties of the alloy compared to CP-Ti. This completely correlates with the results obtained by the XRD experiments. During the welding process, a significant decrease of the β-Ti phase was observed in the volume of the weld seam, which is a result of the high thermal input applied in that area, followed by a subsequent rapid cooling of the weld seam. This leads to the formation of an α'-Ti martensite which has a hexagonal closed-packed crystal structure. This structure is formed via the transformation of the β-Ti bcc phase towards the α'-Ti martensitic one. This transformation was observed due to the addition to the material to a substantial amount of heat followed by very rapid cooling. During the electron-beam processing of the materials, the cooling rate can reach values of $10^4$–$10^5$ K/s, which is the reason for the formation of the martensitic structure. The formation of martensitic phases in α + β titanium alloys depends on the chemical composition and cooling rate. The starting and finishing temperature of the formation of the martensitic structure depends on the amount of the beta stabilizing elements (V in this case), where a larger amount of vanadium leads to a decrease of these temperatures [31–33]. If the finishing temperature is below the room temperature, the beta phase is not completely transformed to the martensitic one. In the case of a lower starting temperature than the room temperature, the discussed transformation is suppressed and the beta phase remains untransformed [31–33]. Previous research indicates that subjecting Ti64 to electron-beam processing results in a transition from a double phase structure of α + β-Ti to the aforementioned α'-Ti martensite [34]. This is in agreement with the data obtained in the present work.

As mentioned previously, the influence of the concentration of the beta stabilizing elements has a direct correlation with the temperature of transformation of the β-Ti phase into the α'-Ti martensitic one. In order to obtain more information regarding the structure and the mechanisms of this transformation during the process of electron beam welding, an energy-dispersive X-ray spectroscopy was performed, and the results are presented in

Table 1. The predominant element in the structure of the specimen welded using a beam power of 1500 W was, unsurprisingly, CP-Ti. The concentration of the V beta stabilizing element is 0.48 wt%,. In the case of the weld formed using a beam power of 2100 W, the concentration of V in the structure of the weld seam is 1.04 wt%,. This is in agreement with the data obtained with the scanning electron spectroscopy experiments that suggest that welding using a beam power of 1500 W leads to the increased transformation of the β-Ti phase towards the α'-Ti martensitic one due to the decrease of the stabilizing elements, vanadium in particular.

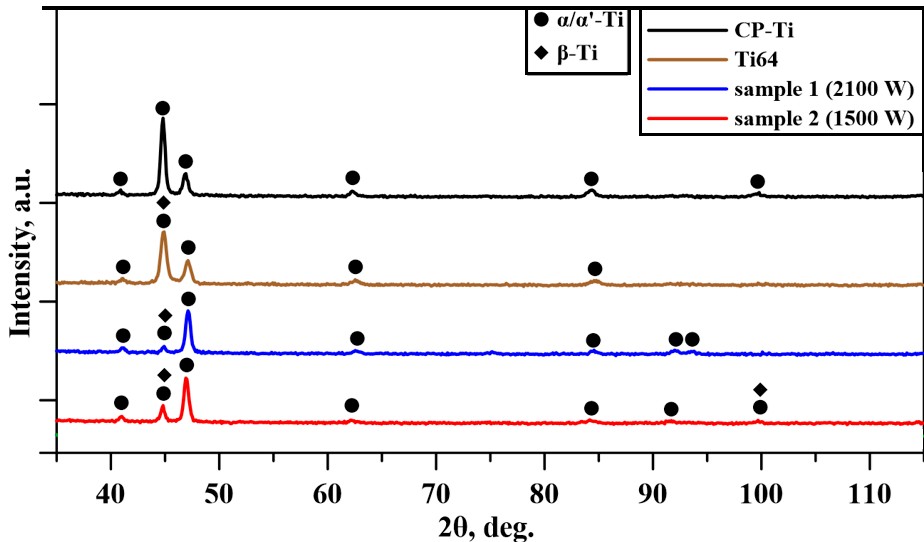

**Figure 6.** X-ray diffraction patterns of the welded joints and the untreated materials.

**Table 1.** Distribution of the elements observed in the weld seam.

| Sample | Distribution of the Elements, wt.% | | |
|---|---|---|---|
| | Ti | Al | V |
| CP-Ti/Ti64 (1500 W) | 97.61 | 1.61 | 0.48 |
| CP-Ti/Ti64 (2100 W) | 96.54 | 2.42 | 1.04 |

In order to evaluate the mechanical characteristics of the welded joints, a series of tensile tests was performed on both the control samples and the electron beam welded ones. Figure 7a shows the prepared tensile test specimens taken from the sample welded using a beam power of 2100 W, and Figure 7b shows the tensile samples of the specimens welded at a power of 1500 W.

Table 2 presents the results of the tensile tests including the yield strength (YS), the ultimate tensile strength (UTS), and the elongation ($\varepsilon$) of both the untreated and the welded materials. The results were obtained by averaging the measured values for each specimen. An average yield strength of 372 MPa, along with an ultimate tensile strength of 511 MPa, was observed in the case of the pure CP-Ti substrate. In the case of the pure Ti64 substrate, a YS of 1031 MPa and a UTS of 1064 was observed. The very close values indicate that the Ti64 substrate has superb elastic deformation properties and high hardness, but low plastic deformation properties. This was confirmed by the elongation of the material observed during the experiments. In comparison to the bulk materials, the welded plates at a beam power of 2100 W and 1500 W have a YS of 125 MPa and 150 MPa, respectively. Their UTS of 510 MPa is almost identical with that of the 2100 W specimen, and that of the 1500 W specimen, which was 502 MPa. The achieved UTS during the experiments is comparable of that of the pure CP-Ti substrate. The YS of the welded plates, however, is significantly lower compared to both bulk materials. As evident by the elongation of the specimens, a

higher plasticity of both specimens was achieved compared to the bulk Ti64 substrate, and lower compared to the CP-Ti one.

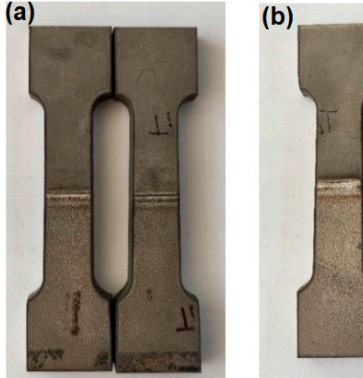

**Figure 7.** Tensile specimens used for analysing the mechanical properties of CP-Ti and Ti64 weld joints welded with a beam power of: (**a**) 2100 W, and (**b**) 1500 W.

**Table 2.** Tensile test results of the untreated welded specimens.

| Specimen | YS, MPa | UTS, MPa | $\varepsilon$, % |
|---|---|---|---|
| CP-Ti | 372 | 511 | 22.1 |
| Ti64 | 1031 | 1064 | 10.6 |
| CP-Ti/Ti64 (2100 W) | 125 | 510 | 17.4 |
| CP-Ti/Ti64 (1500 W) | 150 | 502 | 17.9 |

Figure 8 shows the results of the Vickers hardness measurements. The measurements were performed at the top, middle, and bottom of the weld joint following a trajectory corresponding to the width of the weld seam, starting from one of the substrates and transitioning to the other. Figure 8a presents the results of the experiments for the specimen welded at the highest input power of 2100 W. At the bottom of each specimen the average microhardness of the CP-Ti substrate was initially about 210 $HV_{0.05}$, and slightly increased towards the top of the weld seam to 230 HV0.05. The heat affected zone (HAZ) formed closer to the CP-Ti substrate has an average Vickers hardness of 260 HV0.05 at the bottom of the weld, 320 HV0.05 in the middle, and 295 HV0.05 at the top of the weld. The fusion zone (FZ) has an average microhardness of 370 HV0.05 along the entire cross-section of the weld. The HAZ towards the Ti64 substrate has an initial microhardness of 462 HV0.05 at the bottom of the seam, which decreases to 375 HV0.05 at the middle and top of the weld. The Ti64 substrate has a microhardness of 370 HV0.05 in all cases. Regarding the specimen welded at a beam power of 1500 W (Figure 8b) the microhardness of the CP-Ti substrate was initially 210 HV0.05, and it increased towards the top of the weld to 230 HV0.05. The heat-affected zone closer toward the CP-Ti plate has an average Vickers hardness of 346 HV0.05 at the bottom, 280 HV0.05 in the middle, and 303 HV0.05 at the top. The fusion zone has a microhardness at the bottom and at the top of the weld of 370 HV0.05, and 398 HV0.05 in the middle of the weld. The HAZ towards the Ti64 plate has a value of 375 HV0.05 at the bottom, 390 HV0.05 in the middle, and 401 HV0.05 at the top. The Ti64 plate itself characterizes with a hardness of 360 HV0.05 in all cases. Following the line of the carried out measurements, the microhardness begins to increase from the first heat affected zone formed at the CP-Ti substrate, and increases through the fusion zone, the second heat affected zone, and to the Ti64 substrate. The increase of the microhardness towards the fusion zone can be explained with the increased presence of the formed $\alpha'$-Ti martensitic phase in that area. Similar results were observed by previous researchers [35].

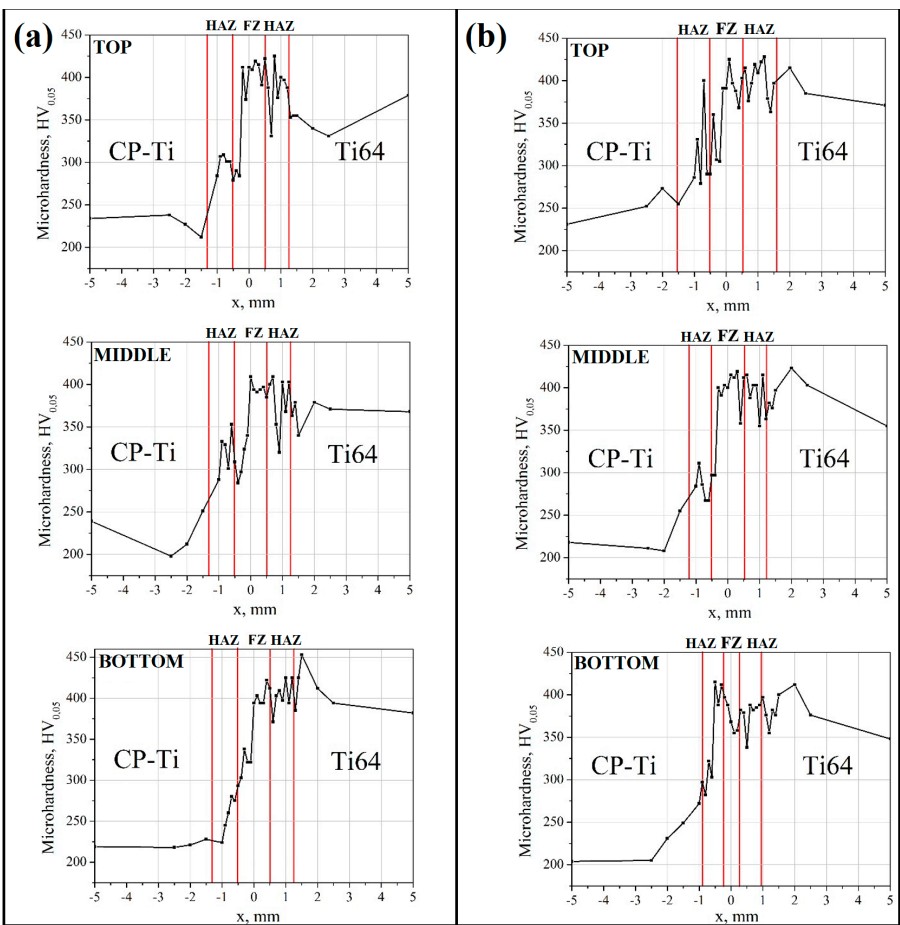

**Figure 8.** Microhardness distribution across the top, middle, and bottom sections of the weld formed using a beam power of: (**a**) 2100 W, and (**b**) 1500 W.

## 4. Discussion

During the experiments, three different power stages of the electron beam were used—2100 W, 1500 W, and 900 W. The size and the shape of the formed weld seam were directly correlated to the input power [36]. In this case, the higher beam power resulted in the full penetration of the welding plates and thus the formation of a symmetrical rectangular shaped weld seam, which lead to the formation of heat affected zones with the same symmetrical shape. Furthermore, the increased power of the electron beam resulted in the formation of narrower heat affected zones. In comparison, the welding plates that were welded with the lower beam power of 1500 W had an irregularly shaped weld seam in the form of a keyhole, which also demonstrates a wider area of the heat-affected zones. In the case of the lowest power of the electron beam, a partial penetration was observed, and a closed keyhole shaped weld seam was formed.

With regard to the phase composition, a noticeable difference between the texturing of substrates and the weld seams was observed. Initially, the substrates had their characteristic composition of a pure α-Ti phase in the case of the CP-Ti substrate, and α-Ti + β-Ti phases in the case of the Ti64 substrate. The X-ray diffraction and the SEM analysis indicated that there was a low amount of the β-Ti phase present in all weld joints. This was caused by the change in the concentration of the beta stabilizing elements in the volume of the weld pool, which induced a transformation of the β phase into the α′-Ti martensitic phase [37]. During the process of welding, the primary α grains along with the α′ phase formed compounds with an elongated structure in the form of thin lamellae [37]. This lead to a refinement of the structure of the weld seam. Evidently, during the process of welding with a beam power of 2100 W, a reduction of the β-Ti phase was observed, although it was accompanied by

an absence of the $\alpha'$ martensitic phase. Generally, where welding processes are concerned, an increase of the heat input in the form of an increase of the heat source's power leads to the slower solidification of the weld seam [38]. This is even more valid for welding processes occurring in vacuum environments, as in the case of electron beam welding. This means that the higher the heat input, the higher the solidification time of the molten pool. Since a 30% difference in power between the 2100 W specimen and the 1500 W specimen was employed, a significantly faster solidification process was observed in the case of the lower powered specimen. Therefore, the thermal cycling gradient in this case is much higher, resulting in the successful transformation of the $\beta$-Ti phase and the formation of the $\alpha + \alpha'$-Ti martensitic phase.

During the process of the welding of both the CP-Ti and Ti64 plates, excellent mechanical characteristics were obtained, as proven by the tensile test and the Vickers hardness measurements. The results of the tensile test experiments indicated that a slight increase of the yield strength of the specimen welded with a power of 1500 W was observed compared to the specimen welded with a beam power of 2100 W. This was undoubtedly caused by the increased formation of the $\alpha'$ martensitic phase, which resulted in the formation of a denser structure. In any case, substantially lower yield strength was observed compared to that of the raw materials. This indicates that the formed weld joint is incredibly susceptible to plastic deformation, as confirmed by the elongation results, which were in both cases higher than that of Ti64, and slightly lower than that of CP-Ti. Similar results were obtained by the authors of [39], who describe the $\alpha'$ phase as highly plastic. Regarding the ultimate tensile strength, both specimens have an almost identical tensile strength that is closer to that of pure titanium and about two-fold lower than that of the titanium alloy. This is of course not surprising considering the structure of the weld seams, particularly in the case of a beam power of 1500 W, which primarily consists of the $\alpha$-Ti and $\alpha'$-Ti phases.

The Vickers hardness is generally influenced directly by the structure of the studied sample. The presence of defects in the structure, particularly in the form of solidification pores, leads to the immediate reduction of the microhardness values. Such pores can occur during the welding process due to the rapid melting and solidification of the material. The fast solidification leads to the shrinkage of the material, which could cause the formation of hollow cavities (pores) in the structure [40]. "Soft" materials with high coefficients of thermal expansion, such as aluminum, are highly susceptible to the incorporation of such defects in their structure [40]. Of course, is the formation of solidification pores is also not uncommon in steels as well, as in the case of the 316L austenitic steel [41]. In the present case, no apparent defects in the structure of all welded seams can be observed in the form of solidification pores or others. The obtained values in the present work indicate that the microhardness increases from the CP-Ti substrate towards the Ti64 one. At the CP-Ti substrate, the initial microhardness is comparable to that of the substrate due to the higher presence of the pure $\alpha$-Ti phase in that area. In the middle of the samples, the microhardness increases to intermediate values between the values of the two substrates. Despite the high presence of the $\alpha$-Ti phase in that area, an increased concentration of the $\alpha'$-Ti phase is present, which is characterized by a slightly denser structure compared to $\alpha$-Ti. The formed martensitic structure hinders the mobility of the dislocations in the structure of the specimens, resulting in the increase of the microhardness. When going toward the Ti64 substrate, the microhardness increases further. The smooth transition of the microhardness indicates that low internal stresses in the structure of the welds were observed, meaning that successful welds with satisfactory mechanical characteristics were formed, as proven by the performed tensile experiments. A noticeable difference in the microhardness of the specimens was observed at the different stages of the cross-section of each weld root. In the case of the weld formed using a beam power of 1500 W, the highest obtained microhardness in the fusion zone was in the middle of the weld. This is attributed to the uneven distribution of the temperature fields in the keyhole shaped weld seam during the welding process. This suggests an inhomogeneous formation and distribution of the different phases obtained during the process, particularly of the $\alpha'$-Ti

martensitic one. In the case of the weld formed using the higher beam power, a significantly uniform formation and distribution of all phases is observed. This is caused by the uniform distribution of the temperature fields attributed to the symmetrical shape of the weld pool formed as a result of the high thermal input. In that case, small symmetrically shaped HAZ were also observed. This leads to the uniform values of the microhardness of that specimen, and the almost perfect separation of the phases based on their location in the substrates. In that case, a predominant concentration of the corresponding to the CP-Ti and Ti64 $\alpha$-Ti and $\alpha$-Ti + $\beta$-Ti phases was observed in both HAZ, respectively. In the case of the substrates welded at a beam power of 1500 W, an irregularly shaped HAZ was observed, as mentioned previously. This is a result of the uneven distribution of the thermal field within the melt pool during the welding process, and subsequently during the cooling process. [42]. This results in the formation of a widening of the weld pool at its top section and the increased formation of the $\alpha'$-Ti martensitic phase. In addition, the higher temperature in that area results in the increased hardening of the substrate materials in the HAZ. This leads to the increase of the microhardness at the top of that specimen and a decrease towards the bottom.

In this work, the possibility of welding CP-Ti with Ti64 was proven. The experiments were carried out at three different power levels of the heat source. In the first case, a larger grain structure was formed, with tiny inclusions of the $\alpha'$-Ti martensitic phase, which led to a slight reduction of the mechanical properties of that specimen compared to the one welded with a lower beam power of 1500 W. The further reduction of the power of the heat source to 900 W led to the formation of a shallow melt pool, and thus to the unsuccessful welding of the two plates. Such a power level can be used to weld thinner weld pieces with a thickness of up to 3 to 4 mm. In the case of the previous two specimens, the formation of a standard "keyhole" weld seam seems to be optimal for welding CP-Ti and Ti64, since this shape of the cross-section of the joint results in the better transitioning of the structure of the substrates along the width of the seam, which leads to an increase of the mechanical characteristics of the specimens.

## 5. Conclusions

Analyzing the results of the current experiments, the following conclusions were derived:

1.  The excessive increase of the beam power leads to the formation of a narrow weld seam with small heat affected zones, and an excessive decrease leads to the formation of a shallow weld pool. The precise adjustment of the power level of the heat source is necessary;
2.  The obtained welded joints are composed primarily of an $\alpha$-Ti phase with small amounts of an $\alpha'$-Ti martensitic phase, where the amount of the $\alpha'$-Ti structure is higher in the weld achieved at a lower power of the electron beam.
3.  The transformation of the beta phase to a martensitic one leads to a smoother transition of the structural properties from one of the substrates to the other, resulting in intermediate microhardness compared to that of the substrates;
4.  The electron beam welding of CP-Ti and Ti64 in all cases leads to a reduction of the yield strength of the materials in the area of the weld seam. The ultimate tensile strength of all seams is comparable to that of CP-Ti.

The results of the current work indicate that electron beam welding is a very promising method for welding CP-Ti and Ti6Al4V, since a high precision of the joint can be achieved with a low number of internal stresses, a low number of defects, and satisfactory mechanical characteristics. The formed joints have moderately high plasticity and good mechanical characteristics, which makes them ideal for applications in specific industrial fields.

**Author Contributions:** Conceptualization, G.K., D.K. and A.A.; methodology, D.K., A.A., S.V., V.D., G.K., B.S. and M.O.; formal analysis, D.K., A.A., S.V., V.D., G.K., B.S. and M.O.; investigation, D.K., A.A., S.V., V.D., G.K., B.S. and M.O.; writing—original draft preparation, G.K., D.K. and S.V.; writing—

review and editing, G.K., D.K. and S.V.; visualization, D.K. and M.O.; project administration, D.K. and A.A. All authors have read and agreed to the published version of the manuscript.

**Funding:** This work was supported by the Bulgarian National Scientific Fund under Grant KP 06-N47/6.

**Data Availability Statement:** Not applicable.

**Acknowledgments:** In memory of our great teacher and scientific supervisor, Peter Petrov, DSc (Institute of Electronics, Bulgarian Academy of Sciences, Sofia, Bulgaria).

**Conflicts of Interest:** The authors declare no conflict of interest.

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
