# Peer review of "Electron-Beam Welding of Titanium and Ti6Al4V Alloy"

_metals, doi:10.3390/met13061065_

Round 1

Reviewer 1 Report

This manuscript presents a comprehensive study on the electron-beam welding process applied to pure α-Ti and Ti6Al4V (Ti64) alloy, addressing the need for research in this area based on a thorough review of existing literature and state-of-the-art technology. The primary focus is to analyze the structure and mechanical properties of the resulting welded joints, with careful consideration of the various power settings of the electron beam. The experimental methodology is well-documented, ensuring reproducibility of the study.

To investigate the phase composition of the welded joints, X-ray diffraction (XRD) experiments were conducted, providing valuable insights. The microstructure analysis was carried out using both optical microscopy and Scanning Electron Microscopy (SEM), enabling a comprehensive examination of the joint characteristics. Tensile tests and microhardness experiments were performed to evaluate the mechanical properties of the weld joints, providing reliable data for analysis.

The obtained results were thoroughly discussed, emphasizing the influence of beam power on the microstructure and mechanical properties of the welded joints. Notably, it was observed that the electron beam welding process led to a reduction in the yield strength of the materials within the weld seam area. However, the ultimate tensile strength of all seams was found to be comparable to that of pure Ti. Furthermore, the practical feasibility of implementing this welding method for α-Ti and Ti64 materials was critically examined.

In conclusion, based on the rigorous experimental investigation and its significant implications for the field of metal joining, this manuscript is expected to attract readers' attention. It is recommended that the authors address the remaining concern outlined by the reviewers before the manuscript can be accepted.

1.     To ensure proper crediting of authors, it is requested to eliminate the use of grouped citations such as [11-15] and [22-28]. It is advisable to limit the number of citations in a single statement to a maximum of three. This adjustment will help maintain appropriate referencing standards throughout the manuscript.

2.     To ensure consistency in notation, it is recommended to maintain uniformity in the representation of terms such as "α-Ti" and "Cp-Ti" throughout the manuscript. By using consistent notation, clarity and accuracy will be upheld, contributing to a more cohesive and professional presentation of the research findings.

3.     In the experimental section, it is advised to focus on describing the methods used rather than specifying the equipment manufacturers, such as Bruker D8 Advance or LYRA3 I XMU. By emphasizing the methodology rather than the specific equipment, the focus remains on the experimental procedures and their reproducibility, enhancing the scientific rigor of the study.

4.     Please annotate the α-Ti particles and α' martensitic phase in Figure 4, similar to how it was done in Figure 3b. Additionally, it would be beneficial to include a discussion explaining the reason for not including the microstructures of the 900 W beam power condition. While briefly mentioned in lines 117-119, it would be beneficial to emphasize and provide further clarity on this decision.

5.     Please incorporate the strain rate that was employed during the tensile testing and provide a rationale for selecting that specific strain rate, supported by relevant references. This addition will enhance the clarity and comprehensibility of the experimental methodology.

6.     Kindly add the line indicating the location of the hardness measurement in Figure 2. Additionally, it would be valuable to discuss the potential impact on the hardness profile if the measurement line were to be shifted towards the top. This consideration will provide insights into the expected changes in hardness distribution and further enrich the interpretation of the results.

Author Response

Dear reviewer,

Thank you so much for your very positive and detailed feedback that will certainly allow us to improve the quality of our manuscript! All mentioned remarks were taken into consideration and the manuscript was edited accordingly.

Once again the author team thanks the respected reviewer for his highly appreciated remarks and recommendations regarding the content of the present paper, that helped us greatly with the process of editing!

Best regards,

Darina Kaisheva

Reviewer 2 Report

The authors conducted a study for joining alpha-Ti and Ti-6Al-4V by using EBW. The results exhibited good mechanical properties (a strength near UTS of Ti and an intermediate elongation). However, there are some issues needed to be addressed before the publication.

1.    In Fig. 4, a microstructure comparison between 2100W and 1500W is shown, and needle-like martensite phase appeared in 1500W, but is seemingly absent in 2100W. Please elaborate on this difference in detail.

2. Further, this could be attributed to different thermal histories in 2100W and 1500W. Please add a section discussing the effect of peak temperature on microstructure differences.

3.  Another characterization method is to use EDS analysis to qualify the elemental distribution locally. It is well known that alloying elements (Al/V) in Ti-64 caused a significant effect on microstructure re-distribution after the welding.

4.  Since XRD only characterize alpha and beta phases, peaks for alpha/alpha-prime cannot be separately found in this profile. So EDS analysis for element re-distribution is required.

5. Overall, a section discussing the relationship between microstructure and mechanical properties is desired.

Author Response

Dear reviewer,

Thank you so much for your very useful and positive remarks regarding our manuscript, that will undoubtedly help us improve the clarity and quality of our manuscript. The appropriate changes in the manuscript were made in agreement with the posted comments.

Once again the author team thanks the respected reviewer for his highly appreciated remarks and recommendations regarding the content of the present paper, that helped us greatly with the process of editing!

Best regards,

Darina Kaisheva

Round 2

Reviewer 1 Report

The authors made significant efforts to address all the reviewers' comments. It is recommended for publication.

Author Response

Once again, the author team would like to express their gratitude for the helpful remarks the respected reviewer provided us with, which resulted in improving the quality of the manuscript!

Best regards,

Darina Kaisheva